# QUANTUM 3D GRAPH STRUCTURE LEARNING WITH APPLICATIONS TO MOLECULE COMPUTING

## ABSTRACT

Graph representation learning has been extensively studied over the last decade, and recent models start to pay attention to a relatively new area i.e. 3D graph learning with 3D spatial position as well as node attributes. Despite the progress, the ability to understand the physical meaning of the 3D topology information is still a bottleneck for existing models. On the other hand, quantum computing is known to be a promising direction for its theoretically verified supremacy for large-scale graph and combinatorial problem as well as the increasing evidence for the availability to physical quantum devices in the near term. In this paper, for the first time to our best knowledge, we propose a quantum 3D embedding ansatz that learns the latent representation of 3D structures from the Hilbert space composed of the Bloch sphere of each qubit. Specifically, the 3D Cartesian coordinates of nodes are converted into rotation and torsion angles and then encode them into the form of qubits. Moreover, Parameterized Quantum Circuit (PQC) is applied to serve as the trainable layers and the output of the PQC is adopted as the final node embedding. Experimental results on two downstream tasks, molecular property prediction and 3D molecular geometries generation, demonstrate the effectiveness of our model. Though the results are still restricted by the computational power on the classic machine, we have shown the capability of our model with very few parameters and the potential to execute on a real quantum device.

## 1 INTRODUCTION

Graph representation, or specifically 3D graph representation as considered in this paper, has received extensive attention over the last decade. Beyond tasks like node classification or link prediction, it further facilitates various downstream applications such as molecular property prediction (Liu et al., 2021) and drug design (Gaudelet et al., 2021). Recently, machine learning approaches have been well developed for learning latent node embedding on molecules (Schütt et al., 2017; Unke & Meuwly, 2019; Gasteiger et al., 2019; 2021). However, the mainstream of such researches is still facing the challenges of better processing the 3D Cartesian coordinates and learning the latent representation of the 3D graph structure.

On the other hand, there are also emerging lines of researches in the area of quantum computing. State-of-the-art quantum computing hardwares are now stepping into the Noisy Intermediate-Scale Quantum (NISQ) era, which leads to the possibility to implement applications in specific scientific domains in the near term (Preskill, 2018; Arute et al., 2019; Zhong et al., 2020; Huang et al., 2020). The overlap between quantum computing and machine learning has emerged as one of the most encouraging areas for quantum computing, as termed by quantum machine learning (Biamonte et al., 2017). Quantum paradigms or hybrid paradigms have been carefully designed to fulfill quantum supremacy in quantum chemistry problems (Aspuru-Guzik et al., 2005; O'Malley et al., 2016). Existing approaches mainly focus on the quantum simulation of molecular energies, which enables effective prediction of chemical reaction rates. However, these quantum approaches (Romero et al., 2018; Peruzzo et al., 2014; O'Malley et al., 2016; Yung et al., 2014) are still simulating the energies of certain small molecules like $H_2$, LiH, etc.

In this paper, we aim to develop quantum machine learning approaches to learn the latent representation of the 3D graph structure of molecules instead of directly simulating the molecular energies with Hamiltonians. Graph learning may not be as precise as molecular simulation approaches for

property prediction, but they have the ability to learn hundreds or thousands of molecules and predict the properties for more complex molecules. Specifically, we first convert the 3D Cartesian coordinates of the atoms into three geometries: distance, rotation angle, and torsion angle. Then we encode the angles and distance as well as the atom type (a discrete variable), into qubits. A distance threshold is used so that each time a focal atom is picked to learn the embedding, one only need to consider the neighboring atoms within the threshold. Considering the size of the molecules and the size of the neighborhood, we only require up to ten qubits to learn the representation, which makes our proposed model easy to simulate on a classical processor and capable of running on a NISQ device. Analog to the hardware efficient ansatz (Kandala et al., 2017; Huang et al., 2021), we apply a Parameterized Quantum Circuit (PQC) after the encoding stage. The trainable parameters are the $\theta$s of the rotation gates $\mathbf{R_x}$ and $\mathbf{R_y}$ in the PQC. The gradient of each parameter $\theta$ is calculated by the shifting technique (Mitarai et al., 2018), and those parameters are updated by the backpropagation and gradient descent approach analog to classical neural networks. We apply a tomography at the end of the circuit and concatenate the real part and imaginary part of the output vector and then take it as the node embedding. We conducted numerical experiments on the filtered QM9 dataset for both molecular property prediction task and molecular geometries generation task. Experimental results show that compared with classical state-of-the-art baseline models, our quantum 3D embedding model achieves comparable results on small datasets with much fewer network parameters. **We summarize our contributions as follows:**

1) To the best of our knowledge, we are the first to use qubits to encode 3D relative positional information, which aims to effectively preserve the property of equivariance and invariance. In fact, using a qubit on a Bloch sphere to encode the rotation and torsion angle of two atoms is more intuitive than using 3D Cartesian coordinates, which is also supported by the success of spherical representation on not only in molecules but also point clouds in recent studies.

2) We use two qubits to represent each atom, and we only consider the focal atom and its neighbors at each iteration. Therefore, the maximum number of qubits is 10 in our model. So we are able to test our model on Qiskit (http://qiskit.org) with quantum cloud service from IBM-Q with simulator yet it guarantees that the code can also be seamlessly deployed and runnable on IBM's NISQ device.

3) We manage to implement a quantum circuit full-amplitude simulator with transition unitary for the PQC on a classical processor. It replicates the results yet over 20 times faster than the QASM simulator from IBM Qiskit's simulator, which enables us to conduct experiments on more tasks.

4) The numerical experiments on two different well-studied molecular tasks show that our embedding approach is able to extract geometry and neighborhood information with very few parameters (only 64 parameters in the PQC) and achieve relatively good results.

## 2 PRELIMINARIES AND RELATED WORKS

In this section, we first briefly review basic concepts of quantum computing as well as quantum machine learning. We further present some previous works on quantum graph learning approaches.

### 2.1 QUANTUM COMPUTING

In quantum computing, qubit (abbreviation of quantum bit) is a key concept which is similar to a classical bit with a binary state. The two possible states for a qubit are the state $|0\rangle$ and $|1\rangle$, which correspond to the state 0 and 1 for a classical bit respectively. We refer the readers to the textbook (Nielsen & Chuang, 2002) for comprehension of quantum information and quantum computing. In this paper, we give a compact description of background for self-containess.

A quantum state is commonly denoted in bracket notation. It is also common to form a linear combinations of states, which we call a superposition: $|\psi\rangle = \alpha|0\rangle + \beta|1\rangle$. Formally, a quantum system on $n$ qubits is an $n$-fold tensor product Hilbert space $\mathcal{H} = (\mathbb{C}^2)^{\otimes d}$ with dimension $2^d$. For any $|\psi\rangle \in \mathcal{H}$, the conjugate transpose $\langle\psi| = |\psi\rangle^\dagger$. The inner product $\langle\psi|\psi\rangle = ||\psi||_2^2$ denotes the square of the 2-norm of $\psi$. The outer product $|\psi\rangle\langle\psi|$ is a rank 2 tensor. Computational basis states are given by $|0\rangle = (1, 0)$, and $|1\rangle = (0, 1)$. The composite basis states are defined by e.g. $|01\rangle = |0\rangle \otimes |1\rangle = (0, 1, 0, 0)$.

Analog to a classical computer, a quantum computer is built from a quantum circuit containing wires and elementary quantum gates to carry around and manipulate the quantum information. A

quantum gate is a unitary operation $\boldsymbol{U}$ on Hilbert space $\mathcal{H}$. When we simulate the quantum circuit on a classical computer, we can obtain the overall transition unitary by tensoring and multiplying those unitary gate operators together.

A projective measurement is described by an observable, $M$, a Hermitian operator on the state space of the system being observed. The observable has a spectral decomposition, $M = \sum_m m \boldsymbol{P}_m$, where $\boldsymbol{P}_m$ is the projector onto the eigenspace of $M$ with eigenvalue $m$. When measuring the state $|\psi\rangle$, the probability of getting results $m$ is given by $p(m) = \langle\psi|\boldsymbol{P}_m|\psi\rangle$.

## 2.2 QUANTUM MACHINE LEARNING

(Cerezo et al., 2021) proposed the concept of Variational Quantum Algorithms (VQA), which leverages quantum advantages to solve machine learning problems on a near-term quantum device. Then, Parameterized Quantum Circuits (PQC) are the concrete implementation of certain VQA. For each qubit we have rotation operator $\mathbf{R_x}(\boldsymbol{\theta})$ which rotate through angle $\boldsymbol{\theta}$ (radias) around the $x$-axis. A PQC is mainly composed of $\mathbf{R_x}(\boldsymbol{\theta})$, $\mathbf{R_y}(\boldsymbol{\theta})$ and $\mathbf{R_z}(\boldsymbol{\theta})$ with $\boldsymbol{\theta}$ as the parameters. The parameters $\boldsymbol{\theta}$ are updated by a classical optimizer to minimize the loss function $\mathcal{L}(\boldsymbol{\theta})$ which evaluates the dissimilarity between the output of PQC and the target result. The derivative of the $i$-th parameter $\boldsymbol{\theta}(i)$ can be computed by using the shifting technique proposed by (Mitarai et al., 2018). It requires running the whole circuit twice but with shifting $\boldsymbol{\theta}(i)$ to $\boldsymbol{\theta}(i) + \pi/2$ and $\boldsymbol{\theta}(i) - \pi/2$

$$\frac{\partial \mathcal{L}(\boldsymbol{\theta})}{\partial \boldsymbol{\theta}(i)} = \frac{\mathcal{L}(\boldsymbol{\theta}(1), \cdots, \boldsymbol{\theta}(i) + \pi/2, \cdots) - \mathcal{L}(\boldsymbol{\theta}(1), \cdots, \boldsymbol{\theta}(i) - \pi/2, \cdots)}{2} \qquad (1)$$

Also using gradient backpropagation, classical learning models are adapted into their quantum version, e.g. QCNN (Cong et al., 2019), QRNN (Bausch, 2020), QGAN (Huang et al., 2021), QLSTM (Chen et al., 2022), and etc, which yet show that the quantum counterparts on NISQ device may not be as powerful as the SOTA classical ones (usually with millions of parameters). Involving quantum computing is an interesting experiment to seek potential supremacy and the connection between latent space and the mystery quantum entanglement.

## 2.3 UNITARY COUPLED-CLUSTER

One of the most promising area to demonstrate the quantum computing supremacy is quantum chemistry. There have been continuous work in this research area and the mainstream of these work is Unitary Coupled-Cluster (UCC) (Romero et al., 2018; Peruzzo et al., 2014; O'Malley et al., 2016; Yung et al., 2014). UCC focused on solving the time-independent Schrödinger equation for molecular system to predict the chemical properties. The coupled-cluster theory is used to obtain the Hamiltonian of a certain molecule and then use Trotter-Suzuki decomposition to approximate the Hamiltonian on a quantum circuit. The parameters in the rotation gates allow us to train for the minimal ground-state energy. This method provides a hierarchy of wave functions that can be prepared on a quantum computer using a polynomial number of gates. It is believed that UCC can provide better accuracy than classical coupled cluster (Wierschke, 1994; Hoffmann & Simons, 1988; Bartlett et al., 1989), which is also regarded as the "gold standard" of quantum chemistry (Bartlett & Musiał, 2007). However, UCC is an unsupervised learning method with no ground truth and can only evolve one molecule at a time since the circuit is uniquely designed for a certain molecule. There are also evidence showing that the number of parameters in UCC might be still too large to allow practical calculations for large molecules.

## 2.4 QUANTUM GRAPH LEARNING

Different from evolving Hamiltonian and solving the Schrödinger equation with the quantum circuit, we also have quantum graph learning approaches trying to learn the latent representation of the vertex and the graph. A hierarchical architecture based on quantum random walks is employed to extract multi-scale properties of the graph (Dernbach et al., 2018). However, it is vague that how to efficiently construct the diffusion matrix from the quantum states generated by the quantum walkers. The information aggregation is performed by the classical system, which further incurs additional expenses as a consequence of the interaction between quantum and classical environment. (Zhang et al., 2019) and (Ai et al., 2022) suggest to exploit the quantum Hilbert space to rebuild the quantum representation of the graph in the quantum state. But the number of qubits to represent a graph with

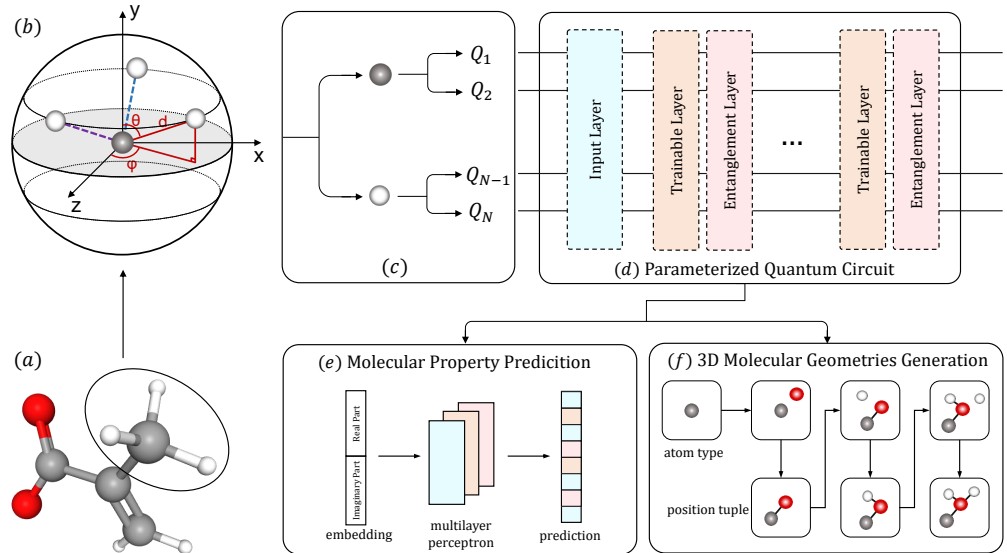

Figure 1: **The quantum 3D embedding scheme.** **(a)** The 3D molecular graph with the gray node (in the black circle) is picked as the focal atom and three white nodes within the distance threshold as the neighbors. **(b)** We convert the 3D Cartesian coordinates of the atoms into the relative position tuple $(d, \boldsymbol{\theta}, \boldsymbol{\varphi})$. **(c)** We encode the position tuple as well as the atom type into two qubits for each atom. **(d)** The PQC for our model, the input layer includes $\mathbf{R_x}$ and $\mathbf{R_y}$ on each qubit, which encodes the up mentioned data. Trainable layers with parameters $\boldsymbol{\theta}$s and entanglement layers are applied alternately to analog the classical machine learning layers. **(e)** The task of property prediction. We use the embeddings from the PQC to predict chemical properties and compare them with the labels. **(f)** The task of 3D molecular geometries generation. We generate a molecule from scratch based on autogressive flow model with picking one focal atom and then deciding the relative position.

its node attributes scales linearly with the number of nodes, and the encoding strategy is not carefully designed. (Bai et al., 2021) and (Chen et al., 2021) develop a hybrid graph learning model which consists of quantum layer and classical layer aimed at reflecting richer graph characteristics. But they both lack formal justifications for the quantum model selections, which lead us to question whether the quantum layer is necessary. Thus, we propose a full quantum paradigm with quantum friendly encoding specially designed for molecular problems.

## 3 METHODOLOGY

### 3.1 PROBLEM SETTING AND METHOD OVERVIEW

**Problem Setting.** In this paper, we aim to develop a quantum machine learning approach for learning node embedding with node-wise 3D coordinates. We take molecules with 3D graph structures as an example. Let $\mathcal{G}$ denotes the graph of a certain molecule and $\mathcal{V}$ denotes the node set of graph $\mathcal{G}$. The number of nodes (in other words atoms) is $n = |\mathcal{V}|$. Each node $v_i \in \mathcal{V}$ has an attribute $a_i$, which is the atom type in our setting. Our target is to learn the embedding for each atom and then obtain the final embedding for the molecule. The embeddings are then tested on different molecular tasks (e.g. molecular property prediction, 3D molecular geometries generation, etc.).

**Method Overview.** We develop a quantum machine learning approach to learn the embedding on 3D graph. The trainable parameter refers to the $\boldsymbol{\theta}$ in those rotation gates in the PQC.

Specifically, we first encode the 3D coordinates and the atom types into qubits. We use relative coordinates instead of the 3D Cartesian coordinates to ensure both equivariance and invariance. The relative coordinates can be written in the form of a position tuple $(d, \boldsymbol{\theta}, \boldsymbol{\varphi})$, where $d, \boldsymbol{\theta}$ and $\boldsymbol{\varphi}$ denote the radial distance, polar angle, and the azimuthal angle, respectively. We set up a distance threshold to pick the neighbors which can interact with the focal atom. A PQC is then used to learn the latent variables and entangle the qubits together. We further apply a tomography at the end of the PQC and then concatenate the real part and the imaginary part. The overall pipeline is shown in Fig. 1.

## 3.2 THE PROPOSED ATOM2QUBIT

Considering a molecule with $n$ atoms, we take it mathematically as a graph $\mathcal{G}$ with $n$ nodes. For each node $v_i$, we have a corresponding attribute $a_i$, which denotes the atom type and a 3D Cartesian coordinate set $\{x_i, y_i, z_i\}$. Without loss of generality, we first pick $v_i$ as the focal atom and learn the embedding of node $v_i$. The distance between $v_i$ and other nodes $v_j \in \mathcal{V}$ is $d_{ij} = \sqrt{(x_j - x_i)^2 + (y_j - y_i)^2 + (z_j - z_i)^2}$. Note that not all of the node pairs in the graph have interaction in the pairs, we set a maximum distance threshold $d_{max}$ as a hyperparameter. So that $v_j \in \mathcal{N}(v_i)$, if $i \neq j$ and $d_{ij} \leq d_{max}$, which means only the nodes $v_j$ with $d_{ij} \leq d_{max}$ are considered as the neighbors of $v_i$. We then need to convert the 3D Cartesian coordinates of $v_j \in \mathcal{N}(v_i)$ into the position tuple $(d_{ij}, \boldsymbol{\theta}_{ij}, \boldsymbol{\varphi}_{ij})$. The definition of rotation angle $\boldsymbol{\theta}$ and torsion angle $\boldsymbol{\varphi}$ are shown in Fig. 1 (b). Now each node $v_j \in \mathcal{N}(v_i)$ can be uniquely defined by $\{a_j, d_{ij}, \boldsymbol{\theta}_{ij}, \boldsymbol{\varphi}_{ij}\}$.

When we encode classical information into the quantum form, we have two different ways. The first one is amplitude encoding and the second one is angle encoding. The amplitude encoding can encode a classical one-hot vector of dimension $n$ with only $\log_2(n)$ qubits, but it is quite hard to encode continuous variables while requires $\mathcal{O}(n)$ times to encode the information. On the contrary, the angle encoding requires a minimum of $n/3$ qubits to encode $n$ classical information, but it is capable of encoding both discrete and continuous variables. Furthermore, the angle encoding is a better fit for the rotation parameters in the circuit. In this paper, we pick angle encoding as our way to encode the information set $\{a_j, d_{ij}, \boldsymbol{\theta}_{ij}, \boldsymbol{\varphi}_{ij}\}$ into qubits.

For each qubit, we have three rotation operators $\mathbf{R_x}$, $\mathbf{R_y}$ and $\mathbf{R_z}$. We can theoretically encode three different pieces of information on one qubit. However, if we consider the qubit on a Bloch sphere, we can uniquely define the rotation track on the Bloch sphere using only two rotation operators. To avoid the decomposition of the third input, we only use two of the rotation operators $\mathbf{R_x}$ and $\mathbf{R_y}$ in this paper ($\mathbf{R_z}$ does not change the outputs of our measurement method). Therefore, we need two qubits $|\Psi_1\rangle$ and $|\Psi_2\rangle$ to encode each node $v_j$,

$$|\Psi_1\rangle = \boldsymbol{U}_{\mathbf{x}}(\boldsymbol{\theta}_{ij}) \times \boldsymbol{U}_{\mathbf{y}}(\boldsymbol{\varphi}_{ij}) \times |0\rangle \tag{2}$$

$$|\Psi_2\rangle = \boldsymbol{U}_{\mathbf{x}}(\frac{d_{ij}}{d_{max}} \times 2\pi) \times \boldsymbol{U}_{\mathbf{y}}(\frac{a_j}{a_{num}} \times 2\pi) \times |0\rangle \tag{3}$$

where $a_{num}$ denotes the number of atoms occurred in the dataset and $a_j$ is an integer $\in [1, a_{num}]$. $|\Psi_1\rangle \otimes |\Psi_2\rangle$ is the quantum encoding state of one node generated from initial state $|00\rangle$. If $n = |v_i \cup \mathcal{N}(v_i)|$, the initial state $|\Psi^0\rangle$ for the PQC in Sec. 3.3 after the Atom2Qubit encoding stage is

$$|\Psi^0\rangle = |\Psi_1\rangle \otimes |\Psi_2\rangle \otimes \cdots \otimes |\Psi_{2n-1}\rangle \otimes |\Psi_{2n}\rangle \tag{4}$$

## 3.3 QUANTUM 3D EMBEDDING ANSATZ

We first discuss the number of qubits we need for our approach on molecule problems. Each time we learn the embedding of node $v_i$, we need to encode the information of $v_i \cup \mathcal{N}(v_i)$ into qubits. Therefore the qubit number is linear with the size of $\mathcal{N}(v_i)$. The interaction between atoms in a molecule is bounded by the bond length between atoms. As the bond length increases, the interaction becomes much weaker, which means we barely have multi-hop message passing in our graph. This gives us the possibility to run the test on an existing near-term quantum device. Therefore, we choose hardware-efficient ansatz that has been proved on a superconducting quantum processor with six fixed-frequency transmon qubits by (Kandala et al., 2017) and a 56-bit superconducting quantum processor *Zuchongzhi* by (Huang et al., 2021).

Analog to classical neural network models, the PQC is constructed by layers and each layer has an identical arrangement of quantum gates. Fig. 2 illustrates the general framework of the quantum 3D embedding ansatz. The overall unitary $\boldsymbol{U}(\boldsymbol{\theta}) = \Pi_{l=1}^{L}(\boldsymbol{U}_{ent}\boldsymbol{U}_l(\boldsymbol{\theta}))$, where $\boldsymbol{U}_{ent}$ is the entanglement layer and $\boldsymbol{U}_l(\boldsymbol{\theta})$ is the $l$-th trainable layer. In particular, we have the $l$-th trainable layer $\boldsymbol{U}_l(\boldsymbol{\theta}) = \bigotimes_{k=1}^{N}(\boldsymbol{U}_{\mathbf{y}}(\boldsymbol{\theta}_{\mathbf{y}}^{(k,l)})) \times \bigotimes_{k=1}^{N}(\boldsymbol{U}_{\mathbf{x}}(\boldsymbol{\theta}_{\mathbf{x}}^{(k,l)}))$, where $\boldsymbol{U}_{\mathbf{x}}$ is the unitary of gate $\mathbf{R_x}$ and $\boldsymbol{\theta}_{\mathbf{x}}^{(k,l)}$ is the parameter for $\mathbf{R_x}$ at the $l$-th layer on the $k$-th qubit. The entanglement layer $\boldsymbol{U}_{ent}$ consists of CNOT gates and

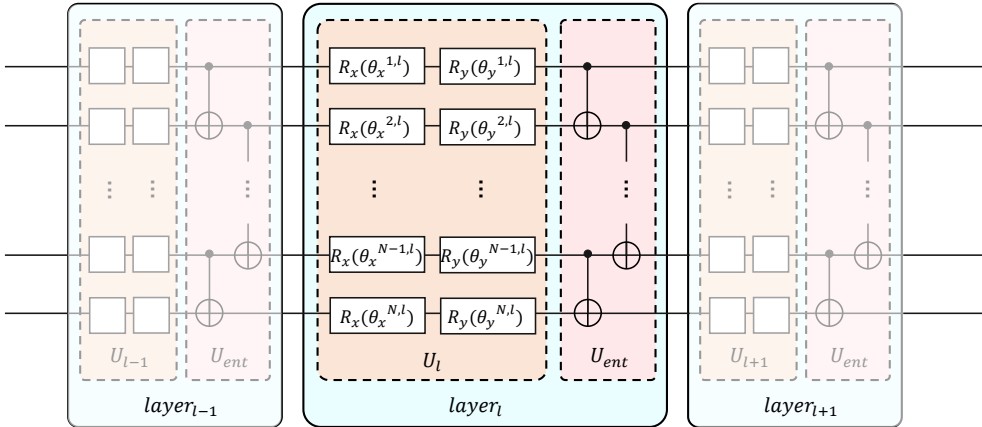

Figure 2: The circuit for our quantum 3D embedding ansatz. Each layer includes trainable parameters block $\boldsymbol{U}_l$ and entanglement block $\boldsymbol{U}_{ent}$. We have $N$ qubits in the circuit so there are $2 \times N$ parameters in each layer. The entanglement layer is composed of CNOT gates to pairwisely entangle all the $N$ qubits.

it entangles all the qubits together shown in Fig. 2. The quantum state $|\Psi^l\rangle$ after $l$ layers is

$$|\Psi^l\rangle = \boldsymbol{U}_{ent} \times \boldsymbol{U}_l \times (\boldsymbol{U}_{ent} \times \boldsymbol{U}_{l-1} \times (\cdots (\boldsymbol{U}_{ent} \times \boldsymbol{U}_1 |\Psi^0\rangle)))) \tag{5}$$

$$= \boldsymbol{U}_{ent} \times \bigotimes_{k=1}^{N}(\boldsymbol{U}_{\mathbf{y}}(\boldsymbol{\theta}_{\mathbf{y}}^{(k,l)})\boldsymbol{U}_{\mathbf{x}}(\boldsymbol{\theta}_{\mathbf{x}}^{(k,l)})) \times (\cdots (\boldsymbol{U}_{ent} \times \bigotimes_{k=1}^{N}(\boldsymbol{U}_{\mathbf{y}}(\boldsymbol{\theta}_{\mathbf{y}}^{(k,1)})\boldsymbol{U}_{\mathbf{x}}(\boldsymbol{\theta}_{\mathbf{x}}^{(k,1)}))|\Psi^0\rangle)))) \tag{6}$$

The quantum state $|\Psi^0\rangle$ is the initial state, which is also the output of the Atom2Qubit stage. With the parameters $\boldsymbol{\theta}_{\mathbf{x}}^{(k,l)}$ and $\boldsymbol{\theta}_{\mathbf{y}}^{(k,l)}$, we can learn the latent representation of each node. Note that the model we proposed is a graph representation learning model, thus we need to further attach downstream tasks to test the efficiency of our model, and the loss function is also obtained from the downstream model. The loss function $\mathcal{L}$ which is employed to optimize the trainable parameters $\boldsymbol{\theta} = \boldsymbol{\theta}_{\mathbf{x}}^{(k,l)} \odot \boldsymbol{\theta}_{\mathbf{y}}^{(k,l)}$, where $\odot$ is concatenation, for our model $\boldsymbol{M}$ it yields:

$$\min_{\boldsymbol{\theta}} \mathcal{L}(\boldsymbol{M}_{\boldsymbol{\theta}}(|\Psi^0\rangle)) \tag{7}$$

The parameters $\boldsymbol{\theta}$ are then updated at each iteration by gradient decent from Eq. 1.

### 3.4 ATOM EMBEDDING

The quantum circuit we mentioned above is a $N$ qubit circuit and it works in a $2^N$ dimensional Hilbert space. We apply a tomography at the end of the circuit thus we can get a $2^N$ dimensional vector with a complex number $\alpha_i$ for each dimension. A quantum state $|\psi\rangle$ can be written in the form of a combination of the computational basis states,

$$|\psi\rangle = \alpha_1 \underbrace{|0\cdots00\rangle}_{2^N} + \alpha_2 \underbrace{|0\cdots01\rangle}_{2^N} + \alpha_3 \underbrace{|0\cdots10\rangle}_{2^N} + \cdots + \alpha_{2^N} \underbrace{|1\cdots11\rangle}_{2^N} \tag{8}$$

where $\alpha_i, 1 \leq i \leq 2^N$ are the complex coefficients and the vector $(\alpha_1, \alpha_2, \cdots, \alpha_{2^N})$ is the result of the tomography. Each $\alpha_i$ can be written in the form of $\alpha_i = \text{Re}(\alpha_i) + \text{Im}(\alpha_i) \cdot i$, where Re denotes the real part, Im denotes the imaginary part and i is the imaginary unit. We then concatenate the real part and the imaginary part of the tomography and get the node embedding vector $(\text{Re}\,\alpha_1, \text{Re}\,\alpha_2, \cdots, \text{Re}\,\alpha_{2^N}, \text{Im}\,\alpha_1, \text{Im}\,\alpha_2, \cdots, \text{Im}\,\alpha_{2^N})$ with the dimension of $2^{N+1}$.

## 4 NUMERICAL EXPERIMENTS

All the experiments are performed on a single machine with 1TB memory, one physical CPU with 28 cores Intel(R) Xeon(R) W-3175X CPU @ 3.10GHz), and two GPUs (Nvidia Quadro RTX 8000). The source code is written by PyTorch, where we simulate the whole quantum circuit process using

transition unitary. We have also implemented a Qiskit version of our modelOn QM9-pred, the average training time for each epoch is over 2 hours using QASM simulator on Qiskit and it takes an average of 310s on our simulator, which is about 23 times faster. Note that all our models are not implemented on quantum hardware yet, but the model and the circuit we proposed are easy to adapt to NISQ devices. To test the performance of our embedding model, we perform numerical experiments on two different tasks and compare the results with state-of-the-art classical 3D molecular representation learning models.

**Dataset**. The benchmark dataset we used is QM9 (Ramakrishnan et al., 2014), which is widely used for predicting various properties of molecules and 3D molecules generating tasks. It includes quantum chemistry structures and properties of up to 134k stable small organic molecules. These molecules consists of up to 9 heavy atoms CONF, not counting hydrogen, and their corresponding 3D molecular geometries are computed by density functional theory (DFT).

Table 1: Statistics of datasets.

| Datasets | QM9-pred | QM9-gen |
|---|---|---|
| #Max Nodes | 10 | 10 |
| #Nodes/Graph | 9.73 | 9.39 |
| #Edges/Graph | 9.37 | 9.16 |

## 4.1 IMPLEMENTATION DETAILS

In Sec. 3.2 we have shown how to convert the information of each atom into rotation angles on qubits. Now we discuss more precisely how to calculate the position tuple $(d, \boldsymbol{\theta}, \boldsymbol{\varphi})$. The 3D coordinates for atom $v_i$ in QM9 is three real numbers $x_i$, $y_i$ and $z_i$. If we pick $v_i$ as the focal atom, $\forall v_j \in \mathcal{N}(v_i)$, we need to calculate the position tuple for atom $v_j$ against $v_i$.

$$d_{ij} = \sqrt{(x_j - x_i)^2 + (y_j - y_i)^2 + (z_j - z_i)^2} \tag{9}$$

$$\boldsymbol{\theta}_{ij} = \arctan(\frac{\sqrt{(x_j - x_i)^2 + (z_j - z_i)^2}}{y_j - y_i}) \tag{10}$$

$$\boldsymbol{\varphi}_{ij} = \arctan\frac{x_j - x_i}{z_j - z_i} \tag{11}$$

In order to fit the definition of rotation angle and torsion angle, those angles should fit into the domain $\boldsymbol{\theta}_{ij} \in [0, \pi]$ and $\boldsymbol{\varphi}_{ij} \in [0, 2\pi)$. We need to adjust the results in Eq. 10 and Eq. 11.

$$\boldsymbol{\theta}_{ij} \leftarrow \boldsymbol{\theta}_{ij} + \pi, \quad \text{if } y_j - y_i < 0 \tag{12}$$

$$\boldsymbol{\theta}_{ij} \leftarrow 0, \quad \text{if } y_j - y_i = 0 \tag{13}$$

$$\boldsymbol{\varphi}_{ij} \leftarrow \boldsymbol{\varphi}_{ij} + \pi, \quad \text{if } z_j - z_i < 0 \wedge (x_j - x_i < 0 \vee x_j - x_i > 0) \tag{14}$$

$$\boldsymbol{\varphi}_{ij} \leftarrow \boldsymbol{\varphi}_{ij} + 2\pi, \quad \text{if } z_j - z_i > 0 \wedge x_j - x_i < 0 \tag{15}$$

Specifically, we set $d_{max} = 1.77$, which is the maximum bond length in the dataset, and we set $a_{num} = 6$ as there are five different types of atom in the dataset in addition with a null type.

Now we discuss more details of the transition unitary based full-amplitude circuit simulation on a classical processor. The quantum gates we used in our PQC are only $\mathbf{R_x}(\boldsymbol{\theta})$, $\mathbf{R_y}(\boldsymbol{\theta})$ and CNOT. The matrix representations of the single-qubit gates are as follows:

$$\mathbf{R_x}(\boldsymbol{\theta}) = \begin{pmatrix} \cos(\frac{\boldsymbol{\theta}}{2}) & -\mathrm{i}\sin(\frac{\boldsymbol{\theta}}{2}) \\ -\mathrm{i}\sin(\frac{\boldsymbol{\theta}}{2}) & \cos(\frac{\boldsymbol{\theta}}{2}) \end{pmatrix}, \quad \mathbf{R_y}(\boldsymbol{\theta}) = \begin{pmatrix} \cos(\frac{\boldsymbol{\theta}}{2}) & -\sin(\frac{\boldsymbol{\theta}}{2}) \\ \sin(\frac{\boldsymbol{\theta}}{2}) & \cos(\frac{\boldsymbol{\theta}}{2}) \end{pmatrix} \tag{16}$$

The two-qubit gate unitary matrix is as follows:

$$CNOT = \begin{pmatrix} 1 & 0 & 0 & 0 \\ 0 & 1 & 0 & 0 \\ 0 & 0 & 0 & 1 \\ 0 & 0 & 1 & 0 \end{pmatrix} \tag{17}$$

With these matrices of the basic gates, we obtain the unitary of a circuit block. We first divide the block by layers, and each layer has at most one gate for each qubit. We tensor the gate matrix within each layer and thus we get the unitary for the layer. We then use matrix multiply between different layers, so the final unitary is calculated through torch.tensor() and then torch.matmul(). For a $N$ qubit circuit, the overall unitary $U \in \mathbb{C}^{2^N \times 2^N}$. The maximum number of qubits we need is 10, so the largest unitary we need is in $\mathbb{C}^{1024 \times 1024}$, which is still affordable on a classical processor.

Table 2: Performance comparison between the baselines and our proposed method on QM9-pred in terms of MAE for three properties ($\epsilon_{HOMO}$, $\epsilon_{LUMO}$ and $\Delta\epsilon$) and the std. MAE for all three properties. We use the unit eV for these three energy-related properties.

| Property | SchNet | | DimeNet++ | | SphereNet | | ComENet | | EGNN | | Ours | |
|---|---|---|---|---|---|---|---|---|---|---|---|---|
| | eV | std.% | eV | std.% | eV | std.% | eV | std.% | eV | std.% | eV | std.% |
| $\epsilon_{HOMO}$ | 0.683 | 90.8 | 0.427 | 56.8 | **0.349** | **46.4** | 0.504 | 67.0 | 0.409 | 54.4 | 0.419 | 55.7 |
| $\epsilon_{LUMO}$ | 0.605 | 45.1 | 0.451 | 33.6 | **0.287** | **21.4** | 0.538 | 40.0 | 0.580 | 43.2 | 0.451 | 33.6 |
| $\Delta\epsilon$ | 0.704 | 60.2 | 0.576 | 49.2 | **0.403** | **34.4** | 0.665 | 56.8 | 0.704 | 60.2 | 0.486 | 41.5 |
| overall | 0.664 | 65.2 | 0.485 | 46.4 | **0.347** | **34.0** | 0.569 | 54.5 | 0.564 | 52.6 | 0.452 | 43.5 |

## 4.2 MOLECULAR PROPERTY PREDICTION

We first conduct experiments on the task of molecular property prediction to evaluate our embedding model. The downstream model we used is a simple multilayer perceptron predictor, which can perform linear regression on the embeddings from the embedding model.

**Setting**. We filter the QM9 to generate the dataset for our prediction task. Our quantum model suffers from the extremely high time cost of simulating the quantum circuit on classic computers, and it is impossible for us to run on the whole 134k molecules in QM9. We sieve the dataset with molecules no more than 10 atoms, and randomly pick 500 of them to form our dataset for the prediction task. We denote the dataset as QM-pred, and statistics of it are listed in Table 1. We split the dataset into training/validation/test sets with a ratio of 8:1:1. Training molecules are used to optimize the model parameters. The validation molecules are used to fine-tune the hyper-parameters as well as conduct the early stopping, and then we report the results on test molecules. Among all sixteen properties listed in QM9, we selected three important energy-related properties, namely $\epsilon_{HOMO}$, $\epsilon_{LUMO}$ and $\Delta\epsilon$. $\Delta\epsilon$, also known as the HOMO-LUMO gap, is one of the most practically-relevant quantum chemical properties of molecules (Bredas, 2014). In line with (Liu et al., 2021), we report the mean absolute error (MAE) for each property as well as the overall mean standardized MAE (std. MAE) for all these three properties.

**Baselines**. To the best of our knowledge, there are no other quantum models considering representation learning for 3D graphs, thus we compare our method with four baselines in the classical domain: the seminal work in this area SchNet (Schütt et al., 2017), DimeNet++ (Klicpera et al., 2020), SphereNet (Liu et al., 2021), ComENet (Wang et al., 2022) and EGNN (Satorras et al., 2021).

**Prediction model**. The embeddings obtained from our model are fed to a simple predictor, which is a multilayer perceptron reducing the size of the embedding from $2^N$ to 1. We use stochastic gradient descent (SGD) with Adam optimizer (Kingma & Ba, 2014) to train our model for a maximum of 100 epochs with a batch size of 32 and a learning rate of 0.01. Meanwhile, as the running time will increase dramatically when the number of trainable layers increases, we set the number of layers in the PQC as four to balance the training time and accuracy.

**Results**. The results of the property prediction task are presented in Table 2. Our model achieved the second best on all three properties. SphereNet is 21.8% better than ours on the overall mean standardized MAE and is 17.1% better than us on the HOMO-LUMO gap. However, SphereNet used $1,898,566$ parameters and we only used $102,881$ parameters (64 of them are from the PQC). Compared to the fundamental baseline, we are 33.3% better than SchNet on overall mean standardized MAE and 31% on HOMO-LUMO gap. Notice that SchNet still used $455,809$ parameters. As shown in Fig. 3, our model converges very fast in the first few epochs, which also demonstrates the efficiency of our model.

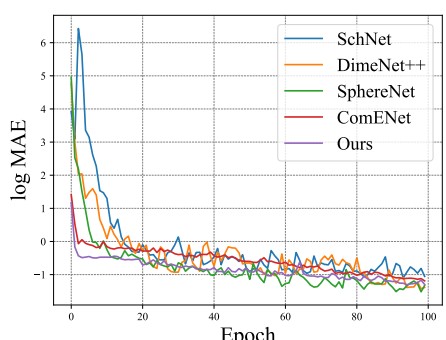

Figure 3: Training loss on the prediction task.

Table 3: Performance of the G-SphereNet and our proposed method on 500 randomly generated molecules for chemical validity percentage and MMD distances of bond length distributions.

| Method | Validity ↑ | MMD distances ↓ | | | | | | |
|---|---|---|---|---|---|---|---|---|
| | | H-C | H-N | H-O | C-C | C-N | C-O | Average |
| G-SphereNet | **68.55%** | **0.161** | **0.280** | 1.104 | 0.399 | 0.438 | **0.277** | 0.443 |
| Ours | 67.00% | 0.237 | 0.409 | **0.770** | **0.326** | **0.407** | 0.378 | **0.421** |

## 4.3 3D MOLECULAR GEOMETRIES GENERATION

This study evaluates the performance of our proposed embedding model when adapted to the existing random molecular geometry generation method. To be more specific, the embeddings from our model are used to extract 3D conditional information in the generation process.

**Setting**. We also use filtered QM9 for evaluation. Different from QM-pred, we select 806 molecules that contain no more than 10 atoms to form our dataset, 50 of them are used for validation and the remaining are used for training. We entitled this filtered QM9 as QM9-gen and the statistics of QM9-gen is presented in Table 1. The generated molecular geometries can be converted to molecular graphs according to the approach proposed in (Gebauer et al., 2019). As for metrics, we use the chemical validity percentage (Validity) which is defined as the percentage of molecular graphs that obey the chemical valency rules to evaluate the generation accuracy. In addition, we adopt Maximum Mean Discrepancy (MMD) (Gretton et al., 2012) distances of bond length distributions to evaluate the 3D structural accuracy of the generated molecular geometries. We calculate the length distribution in the generated geometries and in the dataset geometries separately for each type of bond, then we can obtain the statistical discrepancy between them with the MMD distance. In line with (Luo & Ji, 2022), we compute the MMD on hydrogen-carbon single bonds (**H-C**), hydrogen-nitrogen single bonds (**H-N**), hydrogen-oxygen single bonds (**H-O**), carbon-carbon single bonds (**C-C**), carbon-nitrogen single bonds (**C-N**), carbon-oxygen single bonds (**C-O**) these six types of chemical bonds respectively as they are most frequently appeared.

**Baseline**. We use G-SphereNet (Luo & Ji, 2022) as the baseline in this molecular geometries generation task. We select G-SphereNet (also from ICLR 22) produced by the same group as SphereNet, which uses SphereNet as the embedding model to extract 3D conditional information.

**Generation Model**. As for generation model, we employ the same generation pipeline as G-SphereNet, which adopts a flexible sequential generation strategy by adding atoms in 3D space one by one based on autoregressive flow models. We use Adam optimizer to train the our model for 100 epochs, with a batch size of 64 and a learning rate of 0.001. Also, we set the maximum number of atoms that can be generated for each molecule as 13.

**Results**. We present the performance of our model against G-SphereNet in Table 3. We reach comparable results with baseline model on QM-gen. More specifically, our model slightly outperforms the baseline model on MMD distances for 3 types of bond length, which shows that our method bears a strong capability of extracting the 3D conditional information of molecular geometries.

## 5 CONCLUSION

3D information is important for graphs such as molecules in quantum chemistry and learning the 3D representation for those graphs has attracted increasing attention. Existing classical models face the inherent challenge of understanding the physical meaning of the 3D Cartesian coordinates. To our best knowledge, we are the first to use qubits to encode 3D spatial information and use a Parameterized Quantum Circuit (PQC) to learn the representation of each node as the embedding. The experiments on two well-studied downstream tasks demonstrate the efficiency and capability of our model, and the potential to execute on real quantum devices.

**Limitation & future works.** Our method is limited by the time consumption when simulating quantum circuits, while superconducting NISQ device is entering the 50+ qubit era (Gong et al., 2021), which gives us the confidence to test our model on one of them. Meanwhile, the noise on the gates are not fatal with such shallow circuits. But we will need to adjust the readout procedure of our embedding when testing on NISQ device. It is aimed to extending our experiments to 10 thousand molecules and reaching the chemical accuracy of $1.6 \times 10^{-3}$ Hartree.

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
