# OpenReview forum: "Quantum 3D graph structure learning with applications to molecule computing"
_ICLR.cc/2023/Conference — Submitted to ICLR 2023_

### Official Review · Reviewer_2YoU · 2022-10-21

**Confidence:** 3
**Correctness:** 3
**Technical Novelty And Significance:** 3
**Empirical Novelty And Significance:** 2
**Recommendation:** 6

**Clarity, Quality, Novelty And Reproducibility:**

The paper is in general clearly written and the model is described in detail. The idea of embedding a 3D graph using qubit is novel. However, the competitive edge of quantum computing in the task at hand should be more thoroughly justified through more experiences and ablation study.

**Strength And Weaknesses:**

Strengths:
(1) The paper proposed an interesting methodology and is the first to use qubits to encode 3D relative positional information. It translates the 3D Cartesian coordinates to the relative position tuple (d,θ,ϕ) between the two atoms. Relative coordinates are used to ensure equivariance and invariance.
(2) The paper is clearly written with a coherent structure.

Weaknesses:
(1) In Section 3.3, the authors justify their usage of a distance threshold by saying “As the bond length increases, the interaction becomes much weaker, which means we barely have multi-hop message passing in our graph”. It is known that multi-hop interactions can have non-trivially effects the quantum properties of molecules. It may be interesting to compare the performance of 1-hop vs 2-hop. Section 4, the distance threshold is set to 1.77. Some explanations about this choice of threshold will help. How would a higher distance threshold affect the computing speed and results?
(2) Table 2, why not report the std of each property prediction task? SphereNet performs significantly better than the proposed approach. Is it possible to give some (theoretical) estimations on improvement of performance if more parameters are used in the proposed model?
(3) The idea of using relative coordinates is not novel.


**Summary Of The Paper:**

This manuscript tackles the challenging problem of graph learning in 3D. The authors proposed the first use of quantum computing to learn quantum 3D latent representation of 3D structures from the Hilbert space composed of the Bloch sphere of each qubit. The proposed method preserves equivariance and invariance of 3D graphs. The authors implemented the quantum circuit full-amplitude simulator on a classical processor. The model has a smaller number of parameters compared with existing models but achieved comparable results on small graphs.

**Summary Of The Review:**

This paper present a novel application of quantum computing to learn 3D graphs, which lead to a model with substantially small number of parameters. The lackluster experiment results show that the proposed model works on small graphs. Several weaknesses should be addressed. The proposed method can potentially take advantage of large-scale quantum devices.

---

> ### Author Response · Authors · 2022-11-18
> **Answer to Reviewer 2YoU:**
>
> Your precious comments offer us a lot to explore, which we deeply cherish. We set out below our responses to each of the questions.
>
>
> >***Q1: More discussion about bond length.***
>
> RE: Thank you for your valuable comments. We do realize that 2-hop messages are very important in quantum properties. But 2-hop messages can significantly increase the number of qubits we need. 1-hop is the atoms directly linked by the chemical bond, and the number is closely related to valence. With the maximum degree of 5, we can use 12 qubits to encode the whole group of atoms around the focal atom. If we take 2-hop messages into account, we might need a maximum of 42 qubits to encode the whole neighborhood. We are unable to simulate the quantum circuits over 14 qubits with our classical computing unit. We use the neighbors of a focal atom to learn the embedding, which is similar to the definition of moieties. With the help of type, angles, and distance, we can encode the whole group of atoms around the focal atom into a quantum presentation. The core of our method is more like subgraph matching instead of message passing or GCN, which is commonly used in SE(3)-invariant GNNs. We have to admit that 2-hop messages are also very important to identify a certain moiety, but we currently can not involve 2-hop information. As for the super parameter bond length in the paper, we set it as 1.77 so that all the direct links by the chemical bond are included in the neighborhood.
>
> >***Q2: Std error should be included and require analysis of the results.***
>
> RE: Many thanks for your suggestions. We have updated the script and included std error in the prediction task.
> As stated in [2], quantum machine learning approaches might face the problem of barren plateau with more than 50 layers at 8 qubits, which indicates the existance of deep quantum learning methods. It is known that we can approximate an arbitrary unitary with a group of rotation gates RxRyRx on a single bit, and the rotation gates as long as the two-qubit entangle gates can form a universal gate set. From [3] we can know that approximating an arbitrary unitary operation U on n qubits within a distance $\epsilon$ using $O(n^2 4^n\log^c(n^2 4^n/\epsilon)$ gates. That is more than 23 gates (7 layers) for a 3 qubits circuit to approximate an arbitrary unitary within the distance $\epsilon=0.01$. As for our model, we have a unitrary to map different group of atoms to a quantum state (embedding) that we need to train and fit. According to [3] we can always approximate this unitary with any distance if we have as many gates as possible, and we won't facing the barren plateau issue until we have more than 400 gates.
>
> >***Q3: The idea of using relative coordinates is not novel.***
>
> RE: We admit that using relative coordinates is not novel, but it is also the first time to use qubits to encode the coordinates. This work is not a new approach to improve classical graph learning models on molecular problems. We are writing this paper to shed lights on quantum algorithms solving molecular problems, whose mainstream is UCC (Unitary Coupled-Cluster)[1]. UCC is a Hamiltonian evolution approach, which is an unsupervised learning approach that can only evolve one molecule at a time, and the quantum circuit is uniquely designed for each molecule. Hardware Efficient Ansatz (HEA) can also be used as a substitution for the UCC ansatz, but it is still an unsupervised learning approach. The quantum algorithm we proposed is a supervised learning approach with the ability to learn from thousands of molecules at one time.
>
> [1] Bartlett, R. J.; Kucharski, S. A.; Noga, J. "Alternative coupled-cluster ansatze II. The unitary coupled-cluster method." Chem. Phys. Lett. 1989, 155, 133 – 140.
>
> [2] Jarrod R. McClean, Sergio Boixo, Vadim N. Smelyanskiy, Ryan Babbush, and Hartmut Neven. “Barren plateaus in quantum neural network training landscapes.” Nature Communications, 2018.
>
> [3] Michael A Nielsen and Isaac Chuang. Quantum computation and quantum information, 2002.

---

### Official Review · Reviewer_hAux · 2022-10-24

**Confidence:** 3
**Correctness:** 3
**Technical Novelty And Significance:** 3
**Empirical Novelty And Significance:** 2
**Recommendation:** 5

**Clarity, Quality, Novelty And Reproducibility:**

Generally, this paper is well written. I’m not familiar with quantum computing, but I can get the high-level idea of that through the paper.


**Strength And Weaknesses:**

Strengths:
- This paper introduces using the quantum computing for geometric data modeling. This is a promising direction.
- The paper is self-contained, covering the basic information for quantum ML.

Weaknesses:
- Turn the atom type, distance, and angle into qubits. Then what is the advantage of this operation? The authors claim that this is more “intuitive”, which can be subjective. Can authors provide more objective reason for this?

- Since this work is problem driven, i.e., the main focus is on modeling the quantum chemistry data, then authors may give an overview of this research line. But it is not present in Sec 2. Now the authors give a concrete discussion on quantum ML (sec 2), which is good. But none of them can be used as the baseline (or to be directly related to this problem).

- The insights or motivation of the method is missing. Existing SE(3)-invariant GNNs (DimeNet, SphereNet, GemNet) are using the exact same information for modeling (atom type, distance, and angle), not directly on the 3D Euclidean distance. They can be formulated with the Spherical Fourirer-Bessel basis framework, and this work seems to be using an alternative solution.

- The experimental results are not strong enough to verify the effectiveness of using quantum computing for geometric data modeling. Besides, there are more tasks in QM9 and some key baselines are missing. The baseline compared in this work only covers the SE(3)-invariant NN. There exists another large track of method on SE(3)-equivariant GNNs, such as EGNN[1], Allegro[2].



- A minor point. “However, these approaches are still simulating the energies of certain small molecules like H2…” I think the authors should give reference for this sentence. Actually a lot of work, like NequIP[3], Allegro[2] are considering the MD17 dataset, where the drug molecules are much larger.

[1] Satorras, Vıctor Garcia, Emiel Hoogeboom, and Max Welling. "E (n) equivariant graph neural networks." International conference on machine learning. PMLR, 2021.

[2] Musaelian, Albert, et al. "Learning Local Equivariant Representations for Large-Scale Atomistic Dynamics." arXiv preprint arXiv:2204.05249 (2022).

[3] Batzner, Simon, et al. "E (3)-equivariant graph neural networks for data-efficient and accurate interatomic potentials." Nature communications 13.1 (2022): 1-11.


**Summary Of The Paper:**

This paper provides a novel aspect of using quantum ML for geometric data modeling. This is the first work in this research line, especially for quantum chemistry.

**Summary Of The Review:**

I appreciate the authors for introducing quantum computing for geometric modeling. I think it is a promising direction. I’m familiar with the existing SE(3)-equivariant GNN literature, and from this aspect, the current story is not clear to me because:
- There have been a lot of existing SE(3)- / E(3)-invariant or equivariant GNN models. They are mainly using the spherical harmonic basis (not 3D coordinates), and they have done quite well on the QM9 datasets.
- Empirically, this work is not doing overwhelmingly well on QM9 prediction; and this work is not giving sufficient insights on how this new framework is better than previous work. I would like to raise my score if either of the questions could be solved.

---

> ### Author Response · Authors · 2022-11-18
> **Answer to Reviewer hAux**
>
> We appreciate the reviewer for the thought-provoking comments. We would like to clarify that this work is not a new approach to improve classical graph learning models on molecular problems. We write this paper to shed lights on quantum algorithms solving molecular problems, whose mainstream is UCC (Unitary Coupled Cluster)[1]. UCC is a Hamiltonian evolution approach, which is an unsupervised learning approach that can only evolve one molecule at a time, and the quantum circuit is uniquely designed for each molecule. Hardware Efficient Ansatz (HEA) can also be used as a substitution for the UCC ansatz, but it is still an unsupervised learning approach. The quantum algorithm we proposed is a supervised learning approach with the ability to learn from thousands of molecules at one time. Now we will further respond to the weaknesses one by one.
>
> >***Q1: Why using qubits to encode the atom type, distance, and angle, can the authors provide more objective reason for this?***
>
> RE: Classical algorithms using 3D coordinates can be summarized into two main categories. The first one is directly using them as three real numbers and concatenating them as a new dimension of the atom feature. Another one is using the spherical Bessel function to preprocess the coordinates. However, the spherical wave function or Bessel function is in the form of a Hamiltonian (or to be more specific a Unitary) which is a perfect fit for quantum computing. If we encode the coordinates as well as the distance and atom type into qubits, the circuit can be performed as a trainable Unitary. Setting all the encoding and training processes into a quantum version seems to be more suitable for such a problem.
>
> >***Q2: The authors may give an overview of the research line of modeling quantum chemistry data.***
>
> RE: We have updated the script and involved more related quantum algorithms on molecular problems.
>
> >***Q3: The insights or motivation of the method are missing.***
>
> RE: This work might not be very innovative from a classical NN perspective, but it is a new branch in quantum computing to learn from molecules. It is totally different from the mainstream quantum ml approach UCC and saves us the effort to simulate a Hamiltonian when using Bessel function etc in classical learning. We use the neighbors of a focal atom to learn the embedding, which is similar to the definition of moieties. With the help of type, angles, and distance, we can encode the whole group of atoms around the focal atom into a quantum presentation. The core of our method is more like subgraph matching instead of message passing or GCN.
>
> >***Q4: The experimental results are not strong enough to verify the effectiveness of using quantum computing for geometric data modeling.***
>
> RE: We have updated the experimental part and added the baseline EGNN. We are sorry that we fail to get the code from https://github.com/mir-group/allegro running. There are 12 different tasks in QM9 and most of the approaches show a similar pattern along all 12 tasks. It will take much more time to complete the whole table, and we promise to fill out all the tasks before camera-ready. As for the experimental results, quantum algorithms barely take SOTA classical ml models as their baselines. The ultimate goal for us is for sure the "chemical accuracy" and we hope we can achieve this with a larger training set with more qubits and deeper circuits. Right now we still do not have easy access to quantum devices and the simulation is very time-consuming.
>
> >***Q5: Clarity of a certain sentence.***
>
> RE: The phrase "these approaches" refers to the quantum paradigm UCC. UCC methods require uniquely designed circuits for each molecule and that's why most of them are still working on simple molecules.
>
> [1] Bartlett, R. J.; Kucharski, S. A.; Noga, J. "Alternative coupled-cluster ansatze II. The unitary coupled-cluster method." Chem. Phys. Lett. 1989, 155, 133 – 140.

---

### Official Review · Reviewer_YuG5 · 2022-10-26

**Confidence:** 5
**Clarity, Quality, Novelty And Reproducibility:** The ideas here are clear and straight…
**Correctness:** 3
**Technical Novelty And Significance:** 2
**Empirical Novelty And Significance:** 2
**Recommendation:** 3

**Strength And Weaknesses:**

+ interesting idea of encoding directly the spatial information in the degrees of freedom of a qubit
- very little evidence that this outperforms many other quantum neural network ideas out there
- no theoretical understanding of the architecture. For example, barren plateau will appear in this architecture

**Summary Of The Paper:**

The authors present a quantum neural network architecture for 3D graph learning. the main idea of the paper is to encode the spatial information into a qubit before training the neural network. This idea is quite straightforward and the evidence that this outperforms classical or other quantum architectures is really minimal.

**Summary Of The Review:**

A simple idea for a quantum neural network for 3D structure that misses both theoretical analysis of convergence (barren plateau, etc) and also no siginificant experimental evidence that it is better than other techniques.

---

> ### Author Response · Authors · 2022-11-18
> **Answer to Reviewer YuG5:**
>
> Thanks for your careful and valuable comments. We will explain the concerns point by point.
>
> We are glad that the reviewer brings up the comparison between our method and other quantum neural network ideas. The mainstream of quantum algorithms solving molecular problems is UCC, which is a Hamiltonian evolution approach. However, UCC is an unsupervised learning approach that can only evolve one molecule at a time, and the quantum circuit is uniquely designed for each molecule. Hardware Efficient Ansatz can also be used as a substitution for the UCC ansatz, but it is still an unsupervised learning approach. The quantum algorithm we proposed is a supervised learning approach with the ability to learn from thousands of molecules at one time. We would like to shed lights on quantum approaches that UCC and other Hamiltonian evolution algorithms are not the only ways to solve molecular problems. (It would be very nice if the reviewer could list a few of the quantum neural network approaches that are similar to ours and we will carefully compare them.)
>
> The second concern is our approach lacks a theoretical guarantee that barren plateau might occur in our architecture. We have to admit that barren plateau is an infamous problem that haunts the development of quantum neural networks. However, according to [1], barren plateau will occur when the number of quantum layers is larger than 50 (with the number of qubit at 8), which is much more than the number of layers we use in our experiments (4 layers of rotation and entanglement). Apart from that, the Hardware Efficient Ansatz(HEA) we use to learn the latent embedding is a well-known and well-studied quantum ansatz, which has also been proved on real quantum devices [2,3] (see also section 3.3). HEA to quantum neural network is just like fully-connected layer to the classical neural network. Thus, we omit the theoretical analysis of convergence.
>
> We understand that the reviewer may think this paper is not filled with mathematical derivations. The main contribution of this work is that it shows the possibility to use supervised learning approaches to learn thousands of molecules at one time on a quantum device. The novel encoding of the molecule can help us identify different moieties so that the prediction can be more accurate. As for the experimental results, we are trying to enlarge the training scales, and we hope we can get better results. Quantum machine learning algorithms barely take a fight with the SOTA classical approaches. We list Spherenet from ICLR last year in order to directly show the gap between our method and the SOTA method, and take them as our goal. We hope that people can see that quantum ML is not only about Hamiltonian evolution, but also involves supervised learning approaches with carefully designed encoding method.
>
>
> [1] Jarrod R. McClean, Sergio Boixo, Vadim N. Smelyanskiy, Ryan Babbush, and Hartmut Neven. "Barren plateaus in quantum neural network training landscapes." Nature Communications, 2018.
>
> [2] He-Liang Huang, Yuxuan Du, Ming Gong, Youwei Zhao, Yulin Wu, Chaoyue Wang, Shaowei Li, Futian Liang, Jin Lin, Yu Xu, Rui Yang, Tongliang Liu, Min-Hsiu Hsieh, Hui Deng, Hao Rong, Cheng-Zhi Peng,  Chao-Yang Lu,  Yu-Ao Chen,  Dacheng Tao,  Xiaobo Zhu,  and Jian-Wei Pan. "Experimental quantum generative adversarial networks for image generation." Physical Review
> Applied, 16(2), 2021.
>
> [3] Abhinav  Kandala,  Antonio  Mezzacapo,  Kristan  Temme,  Maika  Takita,  Markus  Brink,  Jerry  M.
> Chow,  and  Jay  M.  Gambetta.    "Hardware-efficient  variational  quantum  eigensolver  for  small
> molecules and quantum magnets." Nature, 549(7671):242–246, sep 2017.

---

### Author Response · Authors · 2022-11-18
**General Response**

Dear Area Chairs and Reviewers,

We appreciate the reviewers' time, valuable comments and constructive suggestions. Overall, the reviewers recognize our work clearly written (YuG5, hAux, 2YoU) and is a promising direction and can potentially take advantages of large-scale quantum devices (hAux, 2YoU).

However, we notice that the reviewers are concerned about the motivation and the experimental results. This work is a full quantum algorithm, and aims to shed lights on quantum machine learning appraches solving molecular problems. The proposed method is totally different from the mainstream quantum approach Unitary Coupled Cluster (UCC), which is a unsupervised learning approach with uniquly designed circuit for each molecule. We tend to use qubits to encode the focal atom as well as its neighbours, which forms a group of atoms similar to the definition of moiety. With this encoding paradigm, we can learn from thousands of molecules at a time, which is novel in quantum molecular learning. As for the experimental results, we run the whole circuit on the full amplitude simulator we implemented in PyTorch, which can only simulate up to 14 qubits. The gradient calculation in quantum machine learning requires repeatedly executing the quantum circuits and this process is very time consuming. We do realize that the results are not comparable to the SOTA classical approaches, but we also barely see quantum machine learning approaches listing SOTA as their baselines. Our ultimate goal is for sure the "chemical accuracy" and we will keep working.

We next provide detailed answers to all the specific questions raised by the reviewers. Further discussions are welcomed to facilitate the reviewing process towards a comprehensive evaluation of our work.

---

### Decision · Program_Chairs · 2023-01-20

**Decision:**

Reject

**Justification For Why Not Higher Score:**

While this work proposes interesting ideas, there are still improvements needed to further show the significant advancement of the proposed method as noted by the reviewers.

**Justification For Why Not Lower Score:**

NA

**Metareview: Summary, Strengths And Weaknesses:**

In this work, the authors propose using quantum computing approach to learn the 3D latent representation of molecular structures, with two solid demonstration in downstream tasks. Overall this work is well presented and reader friendly overall. The major concern is that the overall scope of this work is limited.